# *Lactobacillus rhamnosus* Probio-M9 Improves the Quality of Life in Stressed Adults by Gut Microbiota

**DOI:** 10.3390/foods10102384

**Published:** 2021-10-08

**Authors:** Yan Zheng, Zhongjie Yu, Wenyi Zhang, Tiansong Sun

**Affiliations:** 1Key Laboratory of Dairy Biotechnology and Engineering, Ministry of Education, Inner Mongolia Agricultural University, Hohhot 010018, China; 15147106065@163.com (Y.Z.); wayuzhongjie@163.com (Z.Y.); zhangwenyizi@163.com (W.Z.); 2Key Laboratory of Dairy Products Processing, Ministry of Agriculture and Rural Affairs, Inner Mongolia Agricultural University, Hohhot 010018, China; 3Mongolia Key Laboratory of Dairy Biotechnology and Engineering, Inner Mongolia Agricultural University, Hohhot 010018, China

**Keywords:** *Lactobacillus rhamnosus*, Probio-M9, stressed adults, gut microbiota, metabolites, gut-brain axis, the quality of life

## Abstract

Objective: To evaluate the effect of the probiotic, *Lactobacillus rhamnosus* Probio-M9 (Probio-M9), on the quality of life in stressed adults. Methods: Twelve postgraduate student volunteers were recruited. Six volunteers received oral Probio-M9 for 21 days, while the remaining six received a placebo instead. Fecal samples were collected from the volunteers before and after the intervention. Metagenomic sequencing, nontargeted metabonomics, and quality-of-life follow-up questionnaires were used to evaluate the impact of Probio-M9 consumption on the gut microbiota and life quality of the volunteers. Results: Probio-M9 improved the psychological and physiological quality-of-life symptoms significantly in stressed adults (*p* < 0.05). The probiotic intervention was beneficial in increasing and maintaining the diversity of gut microbiota. The abundance of *Barnesiella* and *Akkermansia* increased in the probiotics group. The feature metabolites of pyridoxamine, dopamine, and 5-hydroxytryptamine (5-HT) were positively correlated with *Barnesiella* and *Akkermansia* levels, which might be why the mental state of the volunteers in the probiotic group improved after taking Probio-M9. Conclusions: We identified that oral Probio-M9 can regulate the stability of gut microbiota and affect the related beneficial metabolites, thereby affecting the quality of life (QoL) of stressed adults. Probio-M9 might improve the psychological and physiological quality of life in stressed adults via the gut-brain axis pathway. The causal relationship should be further explored in future studies.

## 1. Introduction

A mental health survey by the Graduate Students’ Union (GSU) indicates that postgraduate students normally experience severe stress, moderate anxiety, and mild levels of depression [1]. They usually face a social environment that requires the excellent capabilities of stress management and coping [2]. The primary causes of postgraduate stress include academic and society-related pressures [3]. Even though stress symptoms are universal to graduate students, particularly medical graduate students, during the university period, only a few studies have investigated the pressures on graduate students and have attempted to find ways to mitigate the condition. Thus, it would be of interest to explore methods of effective stress management and coping strategies in order to improve the quality of life of postgraduate students and, in turn, increase their work efficiency and performance.

The relationship between gut microbial metabolism and mental health is one of the most intriguing and controversial microbiome research topics [4]. Since birth, intestinal bacteria have exerted a decisive influence on the development and functions of the human gastrointestinal tract, as well as on the immune, neurological, endocrine, and metabolic systems [5]. The intestinal microbial variation has a close relationship with human health and disease, and this association is likely related to bacterial metabolites [6,7]. Many studies have shown a correlation between the changes in microbial communities and pathophysiology, while the relevant clinical research is increasing. The contribution of gut microbes to psychological processes has become the focus of many studies, and researchers are especially interested in the relationship between gut microbes and host behavior. 

The influence of intestinal microorganisms is not limited to the intestinal tract but can also affect the metabolism of the entire body, and even human behavior, via inflammatory and immune responses. Microorganisms play an important role in emotional experiences, particularly in anxiety and depression. In fact, it is the causal effects of bacteria on stress and anxiety that first reminded researchers of the possible effects of bacteria on mental processes [8]. Human emotions, moods, and even expressions, no longer rely solely on the brain, but also depend on the feedback provided by the gut. A growing number of studies have confirmed that intestinal microbes act on the central nervous system (CNS) through the gut-brain axis bidirectional communication system, affecting the metabolism and behavior of the intestinal barrier system. When engaged in interactions to facilitate secretion and absorption, the flora in the body produce many metabolic neurotransmitters and analogs, such as dopamine, acetylcholine, and short-chain fatty acids (SCFAs) [9,10,11]. In turn, these metabolites affect the intestinal barrier and sensory system through the secretion and absorption of the intestine, simultaneously affecting brain function via blood circulation, as well as the intestinal sensory and afferent nerves [5,12]. Changes in human diets and living conditions can cause the type and content of the intestinal microbial metabolites to impact the immune barrier and sensory systems, ultimately affecting quality of life [13]. 

*Lactobacillus rhamnosus* Probio-M9 (Probio-M9) was isolated from the human colostrum in 2007 by Inner Mongolia Agricultural University. Probio-M9 was a potential probiotic strain. In the gastrointestinal fluid tolerance test, Probio-M9 showed good tolerance [14]. Our team’s preliminary experiments have found that the probiotic compound formulation containing Probio-M9 can maintain the steady state of the gut microbiome in the special environment of sailors in long-distance voyages [15]. The effect of a single-component Probio-M9 formulation on the host gut microbiota has not been assessed yet. The current study looked at the effects of Probio-M9 on the gut microbiota and its metabolism from the perspective of metagenomics and metabolomics. It will be used as the preliminary work of a Probio-M9-related experiment. Although most of the available literature associates gut microbiota composition with human health, development, and disease, causal evidence remains sparse [16]. This study aims to improve the gut microbiota via the intake of probiotics, observing the consequent impact on the quality of life of the host. Therefore, adults under learning pressure were selected as the research subjects, and a randomized double-blind placebo-controlled trial was conducted. The study conducted follow-up investigations from both physical and psychological perspectives, combined with metagenomics technology, to examine the impact of gut microbiota on the quality of life of adults. The research results may be used as a theoretical basis for probiotics in maintaining the intestinal microecological balance, enhancing immune function, and improving human quality of life.

## 2. Materials and Methods

### 2.1. Participants

This study recruited twelve healthy graduate students, including six females and six males. All volunteers were master or doctoral students enrolled in a postgraduate program during the trial period. All participants were required to complete an inclusion/exclusion questionnaire, and their demographic information is shown in Table 1. The questionnaire statistics were completed one week before the start of the study, and the list of persons to be included was determined. Inclusion criteria: healthy adult postgraduate students over 20 years old; body mass index (BMI) within the normal range, i.e., 18.5–24.9); and the expression of a willingness to commit to the entire trial period. Exclusion criteria: suffering from chronic respiratory allergies (defined as needing to take allergy drugs daily), severe gastrointestinal diseases, immunosuppressive disease, or having received related treatments within one year prior to the current trial; pregnant or lactating; caught a cold on the first day of the study; received antibiotic treatment within two months before the current study; received vaccination against influenza within one year prior to the current trial; had a history of allergies to any ingredient used in the probiotic/placebo preparation, including milk allergy; had a history of long-term oral corticosteroids administration; took probiotic products regularly (more than three times a week) within three weeks before the current study.

### 2.2. Experimental Design

This study was a 21-day randomized double-blind placebo-controlled intervention trial. Stratified randomization by gender was used for dividing the volunteers into either the probiotic group (Y group, *n* = 6), or the placebo group (A group, *n* = 6). The random assignment of volunteers was based on computerized random number generation by a statistician who did not have direct contact with the volunteers. During the entire trial, no research team member was aware of the number arrangement or the subjects’ group assignment.

During the intervention period, the probiotic group received a daily probiotic preparation consisting of *Lactobacillus rhamnosus* Probio-M9 (Probio-M9: viable bacteria of 2.5×10^10^ CFU/g of probiotic powder, 2 g per sachet). The control group received 2 g of placebo material that was of the same packaging, composition, appearance, and taste as the probiotic powder, but without containing the probiotic strain. Both materials were manufactured by Jinhua Yinhe Biological Technology Co. Ltd., Jinhua, Zhejiang, China, according to ISO9001 and HALAL standards.

The clinical outcomes were measured by changes in physical and mental well-being, recorded by a weekly self-administered questionnaire, as well as through the examination of the subjects’ fecal microbiomes and metabolomes during the course of the trial. The experimental design was approved by the Ethics Committee of the Affiliated Hospital of Inner Mongolia Medical University, China, 24 July 2020 (approval number No. KY 2020010).

### 2.3. Questionnaire Survey

All participants were asked to self-administer a questionnaire at 0 d (start of the trial), 7 d, 14 d, and 21 d (end of the trial) to measure changes in the quality-of-life parameters during the course of the trial. The questionnaire mainly evaluated outcomes from the perspectives of mental and physical well-being [17]. The questionnaire setting was based on “Symptom Checklist 90” (SCL90) [18], the “Self-Rating Anxiety Scale” (SAS) [19], and the “Self-Rating Depression Scale” (SDS) [20], which also included defecation and respiratory symptoms. The questionnaire administered in this work included 164 items, including 40 questions concerning physical state (mainly subjective physical discomfort in the gastrointestinal tract, the respiratory system, and other parts of the body), and 124 questions concerning mental conditions (including feelings, emotions, thoughts, behavior, living habits, and interpersonal relationships, as well as eating and sleeping challenges).

### 2.4. DNA Extraction and Metagenomic Sequencing

Fecal samples were collected from each volunteer at 0 d and 21 d using a disposable fecal sampler, and the collected samples were temporarily stored in a refrigerator at −80 °C for later use. Fecal samples (about 200 mg) were used for DNA extraction by a metagenomic QIAamp Fast DNA Stool Mini Kit (Qiagen, Hilden, Germany). Pipette 1000 uL CTAB lysate was put into a 2.0 mL centrifuge tube, lysozyme was added, an appropriate amount of the sample was added to the lysate, in a 65 °C water bath, and then inverted and mixed several times during this period to fully lyse the sample. Shotgun metagenomic sequencing was performed using an Illumina HiseqXten instrument (Illumina Inc., San Diego, CA, USA). Libraries were constructed by DNA fragments of a 300 bp length; paired-end reads were generated using 150 bp in both forward and reverse directions. The achieved metagenomic reads were processed with the KneadData quality control pipeline (KneadData. http://huttenhower.sph.harvard.edu/kneaddata accessed on 10 February 2021, v0.7.5). Human contaminating reads were removed by Bowtie2 (v2.3.5.1) [21].

### 2.5. Bioinformatics Analysis

The MetaPhlAn2 and HUMAnN2 pipelines were used to analyze the microbial community compositions of high-quality sequences and their capabilities [10]. Furthermore, based on the species information, the Shannon index and the Simpson index were used to evaluate the α diversity index of each sample, while the function and metabolism of the gut microbiota were analyzed using Parallel-META software (Version 3.4.4) channel information.

### 2.6. Fecal Metabolomics

The fecal samples were stored at -80 °C until processed in order to avoid repeated freezing and thawing. One gram of fecal sample was placed in a centrifuge tube, to which 3 mL of a methanol aqueous solution (*v*:*v* = 3:1) was added. This mixture was vortexed at a high speed for 20 s, ultrasonicated for 5 min, incubated for 1 h at −20 °C to precipitate the proteins, and centrifuged at 13,000× *g* for 20 min at 4 °C. The supernatant was transferred to new tubes, and the extracts were dried in a vacuum concentrator without applying heat. Afterward, 600 μL of acetonitrile aqueous (*v*:*v* = 1:1) solution was added to the dried extract for reconstitution. The solution was filtered into a Waters sample bottle using a 0.22 μL organic filter membrane. A quality control (QC) sample was prepared by pooling an equal amount of each sample, and the QC sample was injected three times before the sample run, and every six samples during determination.

Ultra-performance liquid chromatography-quadrupole time-of-flight mass spectrometry (UPLC-Q-TOF-MS^E^) was performed using an ACQUITY H-class UPLC system (Waters Corporation, Milford, MA, USA), with an ACQUITY UPLC HSS T3 column (2.1 mm × 100 mm, 1.8 μm, Waters Corporation, Milford, MA, USA), coupled with a Xevo G2-XS QTOF-MS (Waters Corporation, Milford, MA, USA). Both positive and negative ionization modes were used to enhance the metabolomic coverage. In the ESI^+^ scanning mode, mobile phase A was a 0.1% formic acid-water solution (Sigma-Aldrich, St. Louis, MO, USA) as aqueous phase; mobile phase B was 0.1% formic acid-acetonitrile solution (Thermo Fisher Scientific, Grand Island, NY, USA) as the organic phase. As for the ESI^-^ scanning mode, mobile phase A was 0.1% ammonium hydroxide-water solution (Sigma-Aldrich, St. Louis, MO, USA), and pure acetonitrile was used as mobile phase B. The applied liquid-phase gradient elution program is shown in Table 2. The data acquisition MS^E^ scan ranged from 50 Da to 1200 Da, with a scan time of 0.2 s, followed by a high-energy scan-transfer collision energy ramp from 10 eV to 60 eV. The capillary and taper voltages were 3 kV and 40 kV, respectively, while the source and desolvation temperatures were 120 °C and 500 °C, respectively. The desolvation and cone gas flows were 800 L/h and 50 L/h, respectively. Internal mass calibration was achieved using a solution of leucine enkephalin (0.2 ng/μL) infused at 10 µL/min, and sampled every 30 s with a scan time of 0.3 s. 

The raw UPLC-Q-TOF-MS^E^ data were imported into Progenesis QI software (Waters Corporation, Milford, MA, USA, Version 2.0), which was employed for peak matching, peak alignment, peak extraction, and normalization. The data were processed using Metabo-Analyst 5.0 (Metabo-Analyst. http://www.metaboanalyst.ca accessed on 1 February 2021) for the principal component analysis (PCA), orthogonal partial least squares discriminant analysis (OPLS-DA), and permutation tests. Metabo Analyst online analysis software was used for the statistical metabolic enrichment pathways and functional analyses.

Butyrate concentration was determined by using an Agilent 7890B gas chromatograph, incorporated with an Agilent 5977C mass spectrometer platform (GC-MS, Agilent Technologies, Santa Clara, CA, USA), with chromatographic Agilent HP-5MS capillary GC columns (30 m × 0.25 mm × 0.25 μm, Agilent J & W Scientific, Folsom, CA, USA). The determination was conducted using a method described by Xiaojiao Zheng et al. [13]. The single ion detection scans (SIM) were quantitatively analyzed using Masshunter quantitative analysis software. Each sample was run in triplicate.

### 2.7. Statistical Analysis

The Wilcoxon rank-sum test was used to determine significant differences in scores obtained from the questionnaires between groups. The paired Mann-Whitney and linear discriminant analysis (LDA) effect size (LefSe) tests were used to analyze the significant differences between the bacterial community structures before and after probiotic intervention. The Benjamini–Hochberg method was used to correct the false positive rate of multiple tests, and a corrected *p* < 0.05 was considered significantly different. Differences in bacterial microbiota community structures between samples were visualized by principal coordinates analysis (PCoA) [22]. The corr.test function in the “psych” package was used to perform Spearman’s correlation analysis. Graphic presentations were created by R software (http://www.r-project.org/ accessed on 10 August 2021) and the Metabo-Analyst platform.

### 2.8. Data and Code Availability

All sequencing data from this trial were uploaded to the NCBI SRA (National Center for Biotechnology Information Sequence Read Archive) database, and the bioproject ID was PRJNA713865.

## 3. Results

### 3.1. Quality of Life Questionnaire Analysis

Before the intervention (0 d), all volunteers were investigated at the baseline, after which the placebo and probiotic intervention trials were conducted. Follow-up surveys were conducted at 7 d, 14 d, and 21 d, and questionnaires related to quality of life were completed. Each item on the questionnaire was scored on a scale of one to five, with the score increasing by one for each degree, from mild to severe. Figure 1 presents heatmaps drawn according to the physical and mental symptom scores. 

The WHO defines health as “a state of complete physical, mental and social well-being, and not merely the absence of disease”. Quality of life represents a comprehensive index system used to measure human health, evaluating individual physiological, psychological, and social functions. It also assesses the relationship with the surrounding environment, representing the measurement trend of modern health [10]. A total of 40 items denoting physical symptoms were included in the follow-up questionnaire to this study, including physical scores for somatization, the upper respiratory tract, the gastrointestinal tract, and defecation. At the end of the intervention period, the Y group’s scores were negatively correlated with the intervention time compared with before the intervention. The scores gradually decreased with probiotic intervention, improving the symptoms of the volunteers. The total scores of some volunteers in the A group showed an upward trend. No significant changes were evident after 21 d of placebo intake (*p* > 0.05), while the color of the heatmap became lighter after 21 d of probiotic intervention, indicating that the physical symptoms of the volunteers improved significantly after 21 d of taking probiotics (*p* < 0.05).

Investigating and following up on the mental symptoms revealed that Probio-M9 improved the mental states of the volunteers (*p* < 0.05) but had no significant effect on the mental state of the subjects before and after the administration of the placebo (*p* > 0.05). After 21 d, probiotic intervention substantially improved the mental (*p* < 0.01) and physical states (*p* < 0.05) of the volunteers compared to the placebo. The results indicate that the administration of Probio-M9 significantly improved the physical and mental symptoms of stress in adults, and that increased daily intake of Probio-M9 may substantially enhance their quality of life.

### 3.2. The Effect of the Probiotic Intervention on the Gut Microbiota Composition 

After 21 d of probiotic intervention, the α-diversity analysis revealed an increase in the Shannon Index and the Simpson Index, while the placebo control group showed the opposite trend (Figure 2a,b). 

In order to further evaluate the influence of Probio-M9 on the structure of the gut microbiota, this study used PCoA to evaluate the differences in the structures of the gut microbiota of the volunteers before and after the intervention (Figure 2c,d). The A group showed a distinct dispersion trend, while the Y group exhibited a more even gut microbial structure during the experiment. The results show that the gut microbiota of the volunteers exposed to the 21-d Probio-M9 intervention became more stable. Probiotics can regulate the intestinal microbes of stressed individuals in specific environments, maintaining the stability of the intestinal microorganisms.

### 3.3. Genomic Characteristics of the Gut Microbiome

Illumina Hiseq X Ten metagenomic sequencing was used to obtain a total of 176.24 Gb of high-quality sequences from 24 fecal samples, with an average of 7.34 ± 1.06 Gb sequences per sample. The HUMAnN2 tool was used to analyze the microbial community structure of the high-quality sequences. At Day 0, the gut microbiota of healthy adults mainly consisted of Firmicutes (60.37%), Bacteroidetes (25.72%), Actinobacteria (5.40%), Proteobacteria (4.60%), and Verrucomicrobia (2.36%) at the phylum level (Figure 3a). No significant change was observed in the fecal microbiota between 0 d and 21 d at the phylum level. Nevertheless, the Bacteroidetes and Verrucomicrobia levels in the Y group increased, while the Firmicutes and Actinobacteria levels decreased; therefore, the ratio of Firmicutes/Bacteroidetes (F/B) was decreased. The results of the A group showed an opposite trend. 

The fecal microbiota compositions of the volunteers at the genus and species levels, before and after completing the intervention trial, are shown in Figure 3b and Figure 3d. Here, 89 genera were detected, 15 of which were major genera exceeding 1% (Figure 3b), accounting for 89.11% of the intestinal bacterial gut microbiota. The dominant genera were *Bacteroides* (23.77%), *Eubacterium* (14.26%), *Roseburia* (10.37%), *Escherichia* (7.56%), and *Faecalibacterium* (5.93%). At 21 d, 89 bacterial genera were detected. A total of 18 bacterial genera had a relative content exceeding 1%, accounting for 90.32% of the gut microbiota. LefSe software was used to analyze the bacterial community changes at 21 d. Significant changes were evident for *Ruminococcus* in the A group, and for *Parabacteroides, Paraprevotella, Oscillospiraceae, Veillonella,* and *Bilophila* in the Y group (Figure 3c). Interestingly, the relative abundance of *Bacteroides*, *Prevotella*, *Barnesiella, Akkermansia, Blautia, Veillonella,* and *Desulfovibrio* increased (*p* > 0.05) (Figure 3), and the relative abundance of *Escherchia* (*p* < 0.05) and *Collinsella* (*p* > 0.05) content decreased in the Y group. While the result in the A group was the opposite.

A total of 230 species were detected on Day 0, of which 26 had a relative content exceeding 1%, accounting for 77.20% of the total gut microbiota (Figure 3d). The most prevalent species included *Eubacterium rectale* (9.49%), *Escherichia coli* (6.82%), *Faecalibacterium prausnitzii* (5.69%), *Roseburia inulinivorans* (5.56%), and *Subdoligranulum unclassified* (5.15%). After 21 d of intervention, the relative content of 27 species exceeded 1%, accounting for 73.31% of the total gut microbiota, and mainly included *Eubacterium rectale* (10.22%), *Faecalibacterium prausnitzii* (5.52%), *Roseburia intestinalis* (4.26%), *Roseburia inulinivorans* (4.01%), and *Subdoligranulum unclassified* (3.99%). At the species level, the content of *Clostridium symbiosum* decreased in the A group, while *Coprococcus* species significantly increased in the Y group (*p* < 0.05) (Figure 4). Furthermore, the Y group remained stable at the species level. It is worth noting that *Akkermansia muciniphila* tended to increase after 21 d of intervention with Probio-M9.

### 3.4. Probiotics Regulated the Metabolic Pathways of the Gut Microbiota

The QC samples, 0-d A group, and probiotic metabolism data were used for PCA analysis (Figure 5a). The QC samples collected in positive and negative ion mode showed tight aggregation, indicating excellent stability of the instrument during the assessment. A t-test was performed on the metabolic data of the two groups at 0 d, showing no significant differences in metabolites. In summary, the two groups of volunteers were at the same metabolic level before starting the experiment. The PLS-DA analysis of the metabolic profiles reveals a clear difference between the placebo and probiotics groups after 21 d of probiotic intervention (Figure 5b).

Further statistical analysis showed a total of 178 differential metabolites in the Y and A groups at 21 d (*p* < 0.05, fold change > 2, VIP value > 1). On the basis of the well-established mummichog algorithm, we performed metabolic pathway enrichment analysis using the KEGG database [23]. There were 39 metabolic pathways identified, among which six were significantly changed in the probiotics group compared with the controls (*p* < 0.05, impact > 0). We identified arachidonic acid metabolism as the metabolic pathway of interest, as both groups exhibited lower *p*-values and greater pathway impacts (Figure 6a). Leukotriene B4 and prostaglandin E2 were found in this pathway, metabolized by lecithin. The level of leukotriene B4 in the probiotics group was significantly downregulated, while the level of prostaglandin E2 was upregulated remarkably. For a better understanding of the metabolic differences between the two groups, we also performed functional enrichment analysis. The results reveal that the top three pathways were: arachidonic acid metabolism; ubiquinone, and other terpenoid-quinone biosynthesis; and phenylalanine metabolism (Figure 6b). In the process, we discovered that pathways for fatty acid synthesis and arachidonic acid metabolism increased. Furthermore, the study identified catecholamine, a neurotransmitter that interacts with the gut metabolism to influence chronic stress, from the microbial metabolites in the Y group [24,25]. In addition, pyridoxamine, a class of vitamin B6 compounds, was also detected in the differential metabolites. Compared with the control group, the intake of Probio-M9 can adjust the composition of the human gut microbiota, changing the metabolic glutamate, alanine, aspartate, and aromatic amino acid pathways. The main two metabolic pathways of glutamate, catecholamine biosynthesis, and oleic acid, may be causally related to stress. We also identified that intake of probiotics can increase the level of 5-hydroxytryptamine (serotonin, 5-HT) by regulating the gut microbiota. This could be related to the change in the tryptophan metabolic pathway after taking probiotics.

We noted that butanoate metabolism was one of 39 metabolic pathways. Short-chain fatty acids (SCFAs) are major microbial metabolites produced during anaerobic fermentation in the gut [26]. Among them, butyrate has received particular attention for its important effects on intestinal homeostasis and energy metabolism [27]. Its physiological levels may influence brain function indirectly by regulating immune responses and vagus nerve stimulation [28]. Therefore, we measured butyric acid and isobutyric acid in the feces. The fecal concentrations of butyrate and isobutyrate were quantified GC-MS to evaluate the SCFAs in the fecal samples of the Y and A groups (Figure 7). The results show that the butyrate and isobutyrate levels were higher in the fecal samples from the individuals taking probiotics (*p* > 0.05).

We analyzed the correlation between species and relative abundance and significant fecal metabolites using the Spearman’s correlation coefficient to investigate how probiotics affect the relation of gut microbes and feature metabolites (Figure 8). The results show a significant positive correlation between *Akkermansia muciniphila*, pyridoxamine, and 6-hydroxymelatonin (*r* = 0.632, *p* = 0.028 and *r* = 0.595, *p* =0.041, respectively). Moreover, oleic acid was also positively correlated with *Akkermansia muciniphila* abundance. *Coprococcus* sp. ART55/1 showed a substantial positive correlation with dopamine (*r* = 0.774, *p* = 0.003). 

## 4. Discussion

Among the diverse microbial communities colonized by the human body, gut microbiota significantly affects the health of the host. Examining the human microbiome has highlighted the importance of biological, environmental, psychological, and social systems to health. Stress affects the microbiome over time, eventually forming a link between gut microbes and health. Increasing evidence suggests that the intestinal microbiome influences the CNS via the “microbiology-gut-brain axis”. 

This study conducted a 21-d intervention trial on stressed graduate students using Probio-M9, which increased the diversity of the gut microbiota of the hosts, improving their mental and physical symptoms, and enhancing their quality of life. The results suggest that probiotic intervention can increase and maintain the diversity of gut microbiota, which may be why the volunteers in the Y group displayed improved mental states after taking the Probio-M9 product. The overall decrease in the diversity of the intestinal flora in the A group may be due to the harsh winter environment and the psychological stress of the volunteers during the trial [29]. The Y group may have benefited from Probio-M9, which increased the diversity [30]. Certainly, some of the reasons for the improvement of QOL problems cannot be ruled out as the volunteers’ self-regulation due to the improvement of the external environment, for example, factors such as the gradual end of the term, and the upcoming winter vacation. Nevertheless, the reasons for this phenomenon require further examination.

After the intervention of Probio-M9, the abundance of *Barnesiell* and *Akkermansia* decreased in the A group and increased in the Y group. Intestinal *Barnesiella* is a dietary polyphenol-targeting bacterium that can eliminate harmful bacteria from the gut and produce SCFAs [31]. *Barnesiella* can use fucosyllactose to colonize the intestine to improve the anti-inflammatory ability of the intestine [32], but the level of *Barnesiella* in the placebo group was significantly reduced. The reason for the reduction of *Barnesiella* may be related to external pressures, and the specific reasons need to be further studied. *Akkermansia* is a Gram-negative obligate anaerobic nonmotile nonspore-forming elliptical eubacterium, classified under the phylum Verrucomicrobia [33]. *Akkermansia* can participate in sugar metabolism to produce short-chain fatty acids from mucin and provide energy for goblet cells that produce mucin [34]. In people with a higher abundance of *Akkermansia* in the intestine, the effect of improving insulin resistance is more obvious when an overweight or obese person is treated with a calorie-restricted diet [35]. *Akkermansia* (1.86%) increased in the Y group, and was negatively correlated with obesity, diabetes, inflammation, and metabolic disorders [36]. In addition, several studies have reported that *Akkermansia* can reduce intestinal mucin and increase short-chain fatty acid abundance, protecting the intestines. The intake of *Bifidobacterium* increased the abundance of *Akkermansia*, which was consistent with the findings of this research. Many studies have confirmed that gut microbiota plays a vital role in depression and anxiety [37,38]. The daily human diet may increase or reduce the threat of emotional problems by changing the gut microbiota composition.

The gut microbiota links the CNS and intestinal nervous system by regulating the gut-brain axis, connecting to the brain for emotional regulation, and facilitating a two-way gut-nerve-brain interaction [39]. Probiotic ingestion provides a method for regulating the structure and metabolism of the human gut microbiota, possibly improving emotional stress. Amino groups, primarily responsible for amino acid synthesis in organisms, are formed by glutamic acid and glutamine in the body. The increased amino acid content in the intestinal tract may inhibit neurological degeneration via stress response, such as alleviating Parkinson’s phenotypes [40]. An experiment involving the effect of nicotine on the gut-brain axis in mice revealed that it caused changes in the gut microbiota and glutamate metabolism that affected the mice’s gut-brain axis. [41]. These results provide evidence that Probio-M9 improved the glutamate metabolism of gut microbiota and regulated the brain-intestinal axis. Besides glutamate, additional neurotransmitters and hormones are involved in intestinal signal transduction. A gut-brain circuit was established, via the vagus nerve and the brain, to regulate mood, memory, and other host functions [42], such as aromatic amino acids, dopamine, and 5-HT. The metabolism of aromatic amino acids, such as tyrosine, tryptophan, and leucine, can enhance the CNS and the brain-derived neurotrophic factor (BDNF), improving brain health via the hypothalamic nerve [43]. In addition, pyridoxamine, a class of vitamin B6 compounds, was also detected in the differential metabolites. Higher B vitamins are positively correlated with better physical health, cognitive ability, social roles, social skills, and less fatigue. The level of B vitamins will positively impact the quality of life of the host [44,45,46]. The main two metabolic pathways of glutamate, catecholamine biosynthesis and oleic acid, may be causally related to stress. Oleic acid supplementation can inhibit the quorum-sensing molecule, AI-2, reducing the depressive behavior of mice [47]. Probiotics increased the 5-HT concentration in the intestine after the intervention. It is well known that 5-HT, as a metabolite of tryptophan, is a hormone and excitatory neurotransmitter. It not only promotes intestinal peristalsis [48,49], but also plays an essential role in depression, autism, neurodegeneration, and other diseases [50]. Many studies have shown that 5-HT plays a regulatory role by regulating the hippocampal nervous system. This metabolic result is consistent with the results of tryptophan synthesis (present in >90% of gut microbial genomes), and dopamine synthesis (present in <5% of gut microbial genomes), annotated in the metagenomics of this study. These results indicate that Probio-M9 may potentially become a “mental probiotic” [51,52].

This study of the gut microbiota and metabolites of stressed adults indicates that gut microbiota may deeply influence CNS activities and host behavior, with chemical signaling involved. The physical symptoms and mental states of stressed adults are generally associated with gut bacteria alterations, while oral *Lactobacillus rhamnosus* Probio-M9 could alter stress-related metabolites in the gut and influence pressure behaviors in adults. The gut-brain axis mainly affects human emotions via three pathways, namely, immune regulation pathways, neuroendocrine pathways, and vagus nerve pathways [53]. Subsequent experiments can increase the understanding of blood metabolism and immunology. In fact, the regulation of gut microbiota by probiotics involves far more than simply regulating the gut-brain axis. Complex interactions in the human body increase the need to examine the effects of probiotics on human health. This is highly significant for clarifying the relationship between the microbiome, and stress and anxiety. Therefore, research on probiotic products and probiotic metabolites will provide new ideas for maintaining human health. An in-depth exploration of the complex effect of probiotics on humans may provide insight into host health.

## 5. Conclusions

In the present study, we identified correlations between the composition of the intestinal microbiota in stressed adults and quality of life (QoL). Probio-M9 may improve the psychological and physiological qualities of life in stressed adults via the gut-brain axis pathway. However, the causal relationship needs to be further explored in future studies. In this study, we show that oral Probio-M9 can induce alterations in gut microbiota, including bacterial community compositions and metabolites, in stressed adults. The perturbation of gut microbiota may influence the chemical signaling of gut-brain interactions and further mediate host psychological problems and behaviors. The perturbation of commensal gut microbiota may influence normal gut-brain communication and change host behaviors.

## Figures and Tables

**Figure 1 foods-10-02384-f001:**
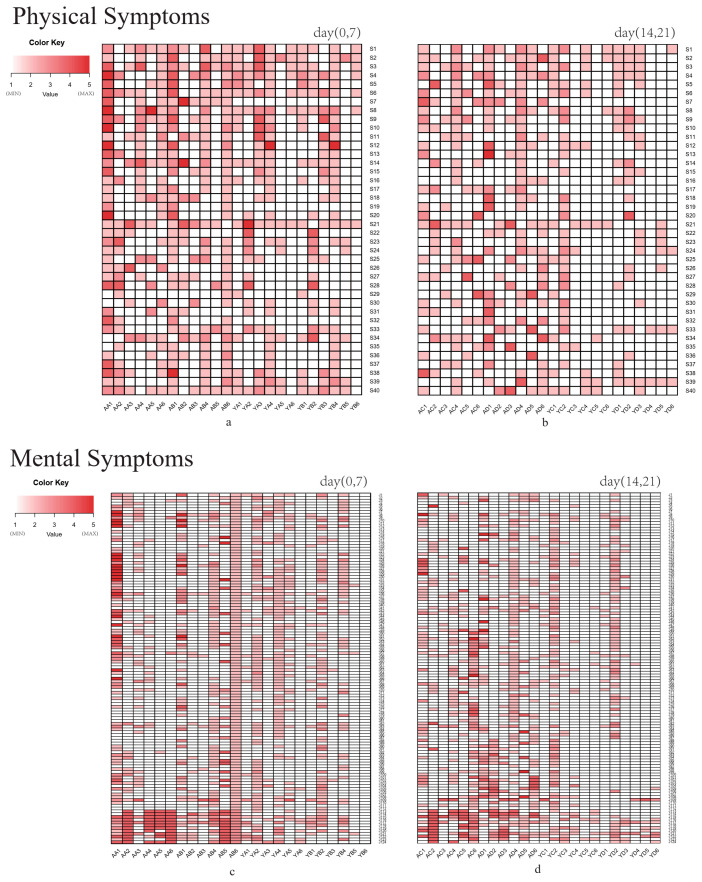
Heatmaps of the self-scaled scores of the physical and mental symptoms for the A and Y groups during different periods. (**a**) The heatmap of the self-scaled scores of the physical symptoms at 0 d and 7 d. (**b**) The heatmap of the self-scaled scores of the physical symptoms at 14 d and 21 d. The abscissa represents the volunteer number, the ordinate denotes the questionnaire item, and the color depth represents the item score. (**c**) The heatmap of the mental symptom self-evaluation list at 0 d and 7 d. (**d**) The heatmap of the mental symptom self-evaluation list at 14 d and 21 d. (Four follow-up surveys were conducted before and during the intervention, which were recorded as A, B, C, and D. For example, the four questionnaire numbers of volunteers A1 and Y1. Their questionnaires are AA1, AB1, AC1, and AD1; and YA1, YB1, YC1, and YD1, respectively. The other questionnaire numbers are the same as above.)

**Figure 2 foods-10-02384-f002:**
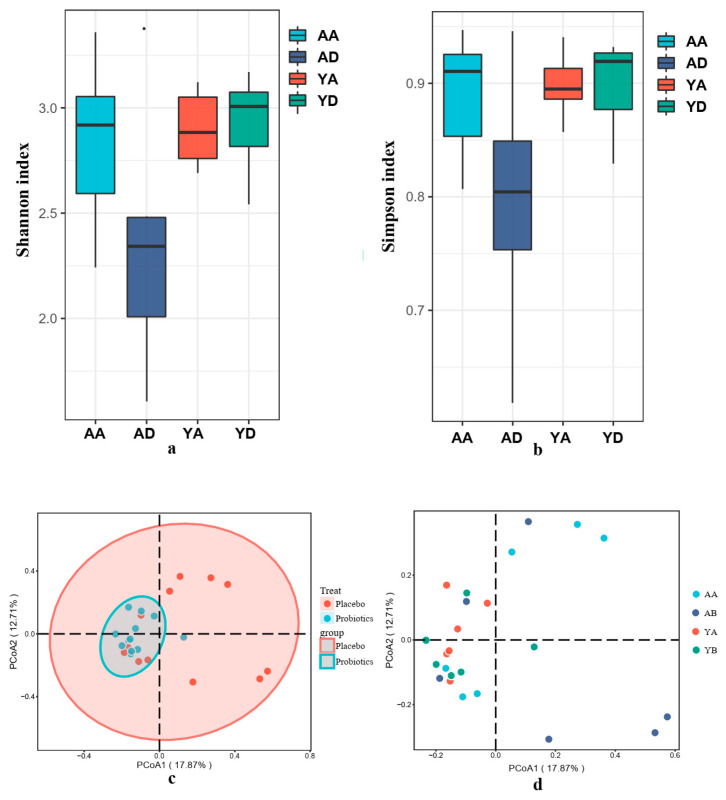
Comparison between the α-diversity indexes and gut microbial population structure of the samples. (**a**) Shannon index; (**b**) Simpson index; (**c**,**d**) The PcoA of the gut microbial population structure. (Figure 2a,b, AA and AD represent the 0-day and 21-day placebo groups, respectively; YA and YD represent the 0-day and 21-day probiotic groups, respectively. The legends of the following results also represent the same group).

**Figure 3 foods-10-02384-f003:**
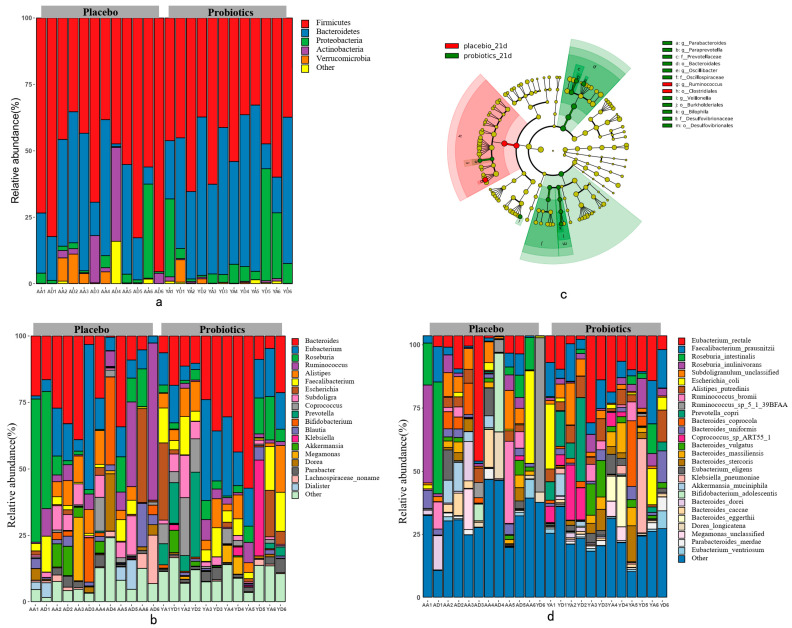
The gut bacterial composition of the volunteers. (**a**) The bar chart of the relative phylum abundance exceeding 1%; (**b**) The bar of the relative genus abundance exceeding 1%; (**c**) 21 d LefSe genus analysis; (**d**) The bar of the relative species abundance exceeding 1%.

**Figure 4 foods-10-02384-f004:**
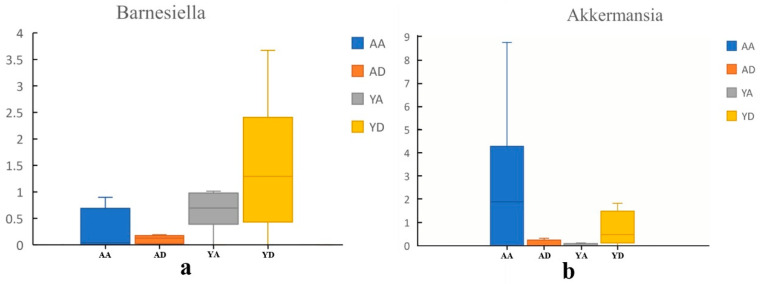
Changes in the abundance of the *Barnesiella* and *Akkermansia* genera. (**a**) Changes in the relative content of *Barnesiella* between the two groups during the experiment; (**b**) Changes in the relative content of *Akkermansia* between the two groups during the experiment (AA and AD represent the 0-day and 21-day placebo groups, respectively; YA and YD represent the 0-day and 6-day probiotic groups, respectively.).

**Figure 5 foods-10-02384-f005:**
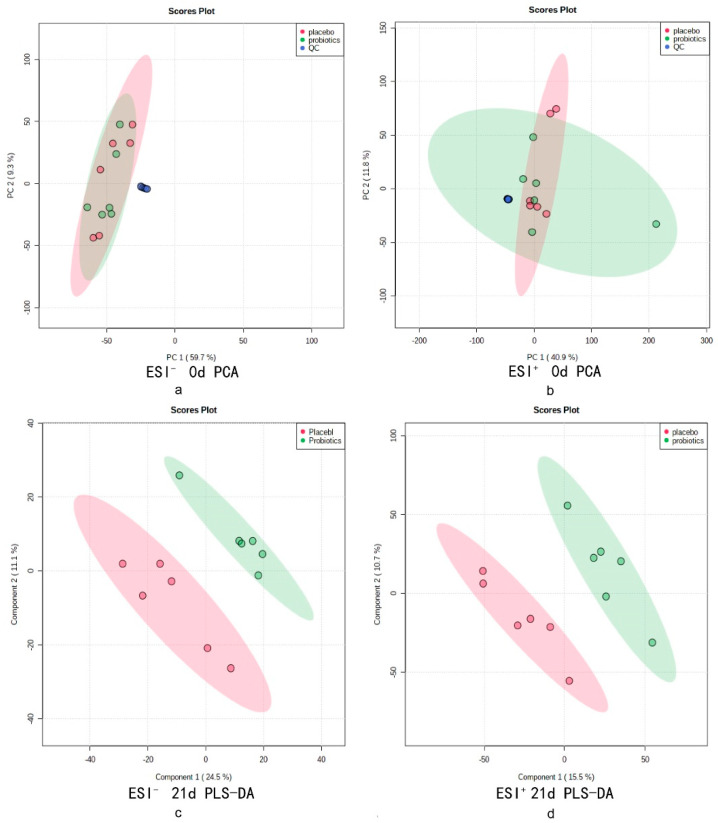
(**a**,**b**) The PCA analysis of QC and metabolic samples in negative and positive ion modes at 0-day; (**c**,**d**) The PLS-DA analysis of metabolic samples in negative and positive ion modes at 21-day.

**Figure 6 foods-10-02384-f006:**
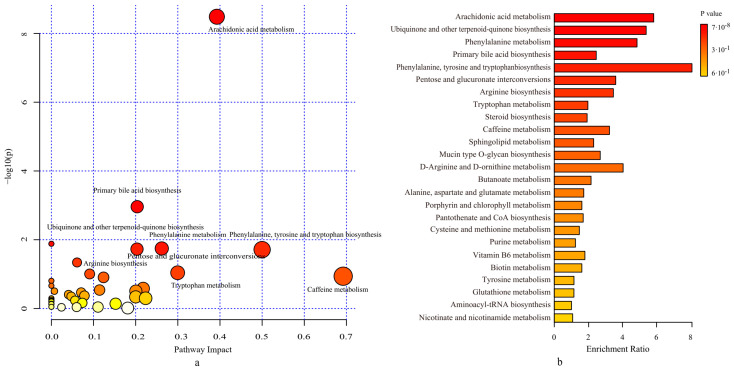
Metabolic pathway analysis of differential fecal metabolites identified from the probiotics group compared with the controls. (**a**) Functional enrichment analysis of pathways using metabolite set enrichment analysis (MSEA). The enrichment ratio is the ratio between observed hits and expected hits; (**b**) Significantly changed pathways based on enrichment analysis (*p*-values) and topology analysis (pathway impact).

**Figure 7 foods-10-02384-f007:**
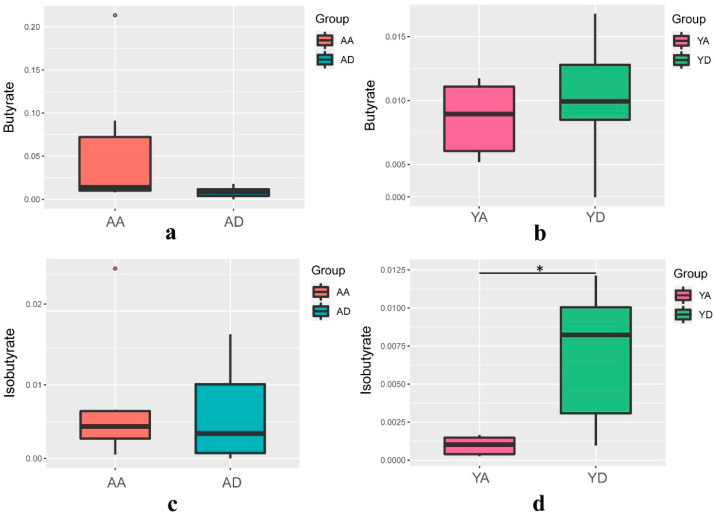
Determination of butyric acid and isobutyric acid in fecal samples during 21 days by GC-MS. (**a**) Changes in butyrate levels in the placebo group; (**b**) Changes in butyrate levels in the probiotics group; (**c**) Changes in isobutyric acid levels in the placebo group; (**d**) Changes in isobutyric acid levels in the probiotics group. (AA and AD represent the 0-day and 21-day placebo groups, respectively; YA and YD represent the 0-day and 6-day probiotic groups, respectively.)

**Figure 8 foods-10-02384-f008:**
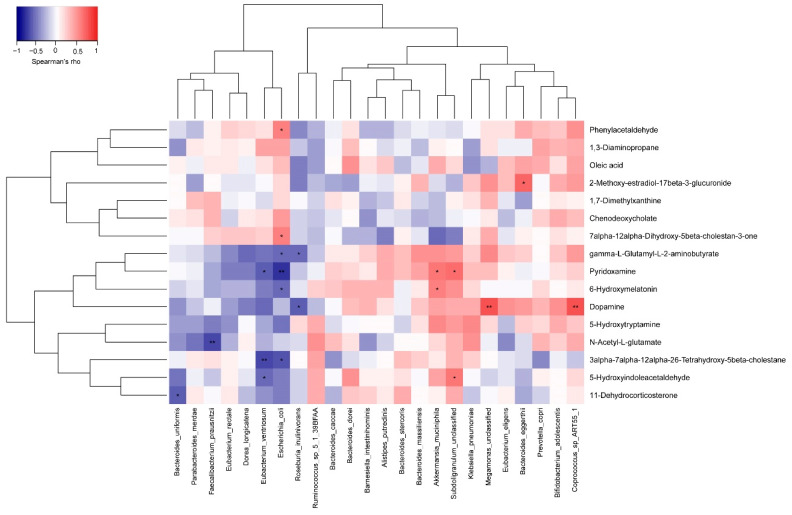
The Spearman’s correlation coefficient for feature metabolites and part of the gut microbes (relative abundance >1%). The heatmap of Spearman’s rank correlation coefficient among 16 feature metabolites and 24 gut microbes with abundances exceeding 0.1%. n = 28; * *p* < 0.05; ** *p* < 0.01; Spearman’s rank correlation.

**Table 1 foods-10-02384-t001:** Subjects’ information.

Group	Subject	Sex	Age	Weight (kg)	Height (cm)	Body Mass Index (kg/m^2^)
Probiotic	A1	male	25	48	152	20.8
	A2	male	26	55	155	22.9
	A3	male	24	49	155	20.4
	A4	female	24	51	166	18.5
	A5	female	26	61	176	19.7
	A6	female	24	81	179	25.3
Placebo	Y1	male	23	87	162	33.2
	Y2	male	26	55	158	22.0
	Y3	male	24	50	162	19.1
	Y4	female	40	65	1.7	22.5
	Y5	female	25	71	185	20.7
	Y6	female	25	71	177	22.7

**Table 2 foods-10-02384-t002:** Phase gradient program.

Gradient Time (min)	Flow Rate (mL/min)	Mobile Phase A (%)	Mobile Phase B (%)	Curve
0	0.32	95.0	5.0	6
2.50	0.32	84.0	16.0	6
12.00	0.32	45.0	55.0	6
15.00	0.32	35.0	65.0	6
18.00	0.32	28.0	72.0	6
20.00	0.32	15.0	85.0	6
25.00	0.32	2.0	98.0	6
28.00	0.32	95.0	5.0	6

## Data Availability

All sequencing data from this trial were uploaded to the NCBI SRA (National Center for Biotechnology Information Sequence Read Archive) database, and the bioproject ID was PRJNA713865.

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
