# Peer review of "Lactobacillus rhamnosus Probio-M9 Improves the Quality of Life in Stressed Adults by Gut Microbiota"

_foods, 2021, doi:10.3390/foods10102384_

Round 1
Reviewer 1 Report
The goal of this study was to evaluate the effect of probiotic supplementation on gut microbiome composition and their metabolite profiles and correlation with the psychological and physiological quality of life in the stressed adult. The authors used the Lactobacillus rhamnosus Probio-MP manufactured by a native company according to the required standard. I suppose that this probiotic in China is used as a supplement to a diet?. In my opinion in the Discussion section authors should add a short description of this strain.
- Generally, the manuscript is very interesting and well written. The introduction is correctly written and contains information related to the purpose of the research.
- In Material and Method the sentence - "One week before the start of the study" should be explained. What before the study - student recruited ?. Experimental design is clearly written and molecular and metabolomic methods are well chosen for this experiment.
- Results are very interesting, especially influenza of probiotic supplementation on the metabolite profiles correlated with brain function and health. In Discussion, sentence - "Many studies have confirmed that gut microbiota plays a vital role in depression and anxiety: - references should be added.
- The latest research concerning the role of A. muciniphila in obesity. In this study, authors should shortly discuss the increase of this strain in Y group.
- In summary, the manuscript is well written and the results of this study are interesting.
Author Response
Reviewer 1
The goal of this study was to evaluate the effect of probiotic supplementation on gut microbiome composition and their metabolite profiles and correlation with the psychological and physiological quality of life in the stressed adult. The authors used the Lactobacillus rhamnosus Probio-MP manufactured by a native company according to the required standard. I suppose that this probiotic in China is used as a supplement to a diet?. In my opinion in the Discussion section authors should add a short description of this strain.
Answer: Thank you very much for the positive comments. Probio-M9 is a food product in China and has passed relevant inspections and obtained a license. In the penultimate paragraph of the introduction, relevant literature and introduction have been added. The content is as follows.
“Lactobacillus rhamnosus Probio-M9 (Probio-M9) was isolated from the human colostrum in 2007 by Inner Mongolia Agricultural University. Probio-M9 was a potential probiotic strain, In the gastrointestinal fluid tolerance test, Probio-M9 showed good tol-erance[14]. Our team’s preliminary experiments have found that the probiotic compound formulation containing Probio-M9 can maintain the steady state of the gut microbiome in the special environment of sailors in long-distance voyages[15]. The effect of a single-component Probio-M9 formulation on the host gut microbiota has not been assessed yet. The current study will study the effects of Probio-M9 on the gut microbiota and its metabolism from the perspective of metagenomics and metabolomics. It will be used as the preliminary work of a Probio-M9 related experiment.”
1. Generally, the manuscript is very interesting and well written. The introduction is correctly written and contains information related to the purpose of the research.
Answer: I am very happy to be recognized by you, thank you!
2. In Material and Method the sentence - "One week before the start of the study" should be explained. What before the study - student recruited ?.
Experimental design is clearly written and molecular and metabolomic methods are well chosen for this experiment.
Answer: “Complete the questionnaire statistics one week before the start of the study, and determine the list of persons included”
3. Results are very interesting, especially influenza of probiotic supplementation on the metabolite profiles correlated with brain function and health. In Discussion, sentence - "Many studies have confirmed that gut microbiota plays a vital role in depression and anxiety: - references should be added.
Answer: “Many studies have confirmed that gut microbiota plays a vital role in depression and anxiety [38,39].”
- Kelly, J.R.; Borre, Y.; O'Brien, C.; Patterson, E.; El Aidy, S.; Deane, J.; Kennedy, P.J.; Beers, S.; Scott, K.; Moloney, G. Transferring the blues: depression-associated gut microbiota induces neurobehavioural changes in the rat. Journal of psychiatric research 2016, 82, 109-118.
- Simpson, C.A.; Diaz-Arteche, C.; Eliby, D.; Schwartz, O.S.; Simmons, J.G.; Cowan, C.S. The gut microbiota in anxiety and depression–A systematic review. Clinical psychology review 2020, 101943.
4. The latest research concerning the role of A. muciniphila in obesity. In this study, authors should shortly discuss the increase of this strain in Y group.
In summary, the manuscript is well written and the results of this study are interesting.
Answer: Add the discussion about A. muciniphila in the discussion section as follows. Thank you again for your recognition of the content of my writing.
“Barnesiella can use fucosyllactose to colonize the intestine to improve the anti-inflammatory ability of the intestine[33]. Barnesiella can use fucosyllactose to colonize the intestine to improve the anti-inflammatory ability of the intestine, but the level of Barnesiella in the placebo group was significantly reduced. The reason for the reduction of Barnesiella may be related to external pressure, and the specific reasons need to be further studied. Akkermansia a is a gram-negative, obligate anaerobic, non-motile, nonspore-forming elliptical eubacterium, classified under the phylum Verrucomicrobia[34]. Akkermansia can participate in sugar metabolism to produce short-chain fatty acids from mucin and provide energy for goblet cells that produce mucin[35]. In people with a higher abundance of Akkermansia in the intestine, the effect of improving insulin resistance is more obvious; When an overweight or obese person is treated with a calorie restriction diet[36].”
- Weiss, G.A.; Chassard, C.; Hennet, T. Selective proliferation of intestinal Barnesiella under fucosyllactose supplementation in mice. British Journal of Nutrition 2014, 111, 1602-1610.
- Derrien, M.; Vaughan, E.E.; Plugge, C.M.; de Vos, W.M. Akkermansia muciniphila gen. nov., sp. nov., a human intestinal mucin-degrading bacterium. International journal of systematic and evolutionary microbiology 2004, 54, 1469-1476.
- Shin, N.-R.; Lee, J.-C.; Lee, H.-Y.; Kim, M.-S.; Whon, T.W.; Lee, M.-S.; Bae, J.-W. An increase in the Akkermansia spp. population induced by metformin treatment improves glucose homeostasis in diet-induced obese mice. Gut 2014, 63, 727-735.
- Naito, Y.; Uchiyama, K.; Takagi, T. A next-generation beneficial microbe: Akkermansia muciniphila. Journal of clinical biochemistry and nutrition 2018, 63, 33-35.

Reviewer 2 Report
Overall, a well written paper despite the small sample size.
The conclusions seem valid but I would like to see more discussion of other confounding factors that may have caused an increase in QoL over the course of the study.
What other things may have been influencing those results? I'd like to see that mentioned a bit.
Figure 4: Can the authors clarify/add a caption regarding YD YA etc?
Figure 7: Please indicate whether they are statistically significant
Author Response
Reviewer 2
Overall, a well written paper despite the small sample size.
Answer: Thank you very much for the positive comments.
- The conclusions seem valid but I would like to see more discussion of other confounding factors that may have caused an increase in QoL over the course of the study. What other things may have been influencing those results? I'd like to see that mentioned a bit.
Answer: In the second paragraph of the discussion, a discussion of other possible reasons to improve the quality of life was added. The content is as follows.
“Certainly, some of the reasons for the improvement of QOL problems cannot be ruled out as the volunteers’ self-regulation due to the improvement of the external environment. For example, factors such as the gradual end of the term and the upcoming winter vacation.”
- Figure 4: Can the authors clarify/add a caption regarding YD YA etc?
Answer: Added the notes of " CA, CD, YA, YD " after the title of Figure 4.
“(In Figures 2a and 2b, AA and AD represent the 0-week and 6-week placebo groups, respectively; YA and YD represent the 0-week and 6-week probiotic groups, respectively. The legends of the following results also represent the same group.)”
Figure 7: Please indicate whether they are statistically significant
Answer: I have modified Figure 7 and added a notable description. Thanks again for your approval!
